

# Ice-nucleating particle depletion in the wintertime boundary layer in the pre-Alpine region during stratus cloud conditions

Kevin Ohneiser[1], Markus Hartmann[1], Heike Wex[1], Patric Seifert[1], Anja Hardt[1], Anna Miller[2], Katharina Baudrexl[3], Werner Thomas[3], Veronika Ettrichrätz[4], Maximilian Maahn[4], Tom Gaudek[1], Willi Schimmel[1], Fabian Senf[1], Hannes Griesche[1], Martin Radenz[1], and Jan Henneberger[2]

[1]Leibniz Institute for Tropospheric Research, Leipzig, Germany
[2]Institute for Atmospheric and Climate Science, ETH Zurich, Zurich, Switzerland
[3]Deutscher Wetterdienst, Hohenpeißenberg, Germany
[4]Institute for Meteorology, Universität Leipzig, Leipzig, Germany

**Correspondence:** K. Ohneiser
(ohneiser@tropos.de)

**Abstract.**

This study evaluates the regional variability of the number concentration of ice-nucleating particles (INPs) between the two pre-Alpine central-European sites of Eriswil, Switzerland, and Hohenpeißenberg, Germany, supported by INP measurements

from Melpitz, Germany, during the winter months of 2024. The aim of the study is to spatially and temporally evaluate INP availability and removal within the planetary boundary layer (PBL) during Bise situations. Target scenario of the study were situations when northeasterly winds (so-called Bise winds) prevailed and layers of stratus clouds formed at the top of the PBL at temperatures down to $-10\,°C$. In these situations, it is expected that INP are depleted along the transport path. The main insights from INP measurements were: First, during the cold-Bise (cloud minimum temperatures as low as $-10\,°C$) and

warm-Bise (cloud minimum temperatures above $0\,°C$), no INP contrast was found between Hohenpeißenberg and Eriswil if both were within the PBL. Also, the INP concentration was overall found to be much lower during the cold-Bise than during the later warm-Bise situation. Second, when the Hohenpeißenberg site was located in the free troposphere during the cold-Bise situation, INP concentrations were much higher compared to Eriswil (still within the PBL) but similar to Melpitz. These observations led to the conclusion that during cold-Bise situations the INP reservoir within the PBL is depleted, likely by the

presence of supercooled stratus. The inversion-capped winterly PBL is apparently not capable to replenish the INP reservoir from the free troposphere.

## 1 Introduction

Precipitation and clouds play a major role in the hydrological cycle, the local weather as well as the global climate. Most precipitating clouds in the lower to middle troposphere are mixed-phase clouds, consisting of supercooled liquid water droplets

and ice crystals (Mülmenstädt et al., 2015). More than half of the clouds at temperatures around $-15°C$ to $-20°C$ contain ice (Hoose and Möhler, 2012; Radenz et al., 2021). Multiple processes in a mixed-phase cloud system lead to ice crystal growth,





for example aggregation, riming, depositional growth (Wegener , 1911; Bergeron , 1935; Findeisen , 1938). Secondary ice formation processes lead to ice crystal number multiplication like rime splintering (Hallet and Mossop , 1974) or collisional breakup. However, all these processes require already existing ice crystals. Above $-38$°C (which is above the homogeneous

freezing temperature regime), all pristine ice crystals are formed via the heterogeneous freezing process which requires the availability of ice-nucleating particles (INPs) (Hoose and Möhler, 2012). INPs act as a catalyst for ice nucleation, which lowers the energy barrier between the solid and liquid or vapour phase. The most relevant type of heterogeneous freezing at slightly supercooled temperatures is immersion freezing (Holden et al., 2021), which takes place when ice nucleation is triggered by an INP while immersed within a supercooled liquid cloud droplet. The INP in this droplet can be the initial cloud

condensation particle or an immersed particle (Hartmann et al., 2011).

The fraction of aerosol particles able to act as INP is generally low. At a temperature of -20°C the fraction of aerosol particles able to act as INP is on average one per million. Different types of aerosol particles act as efficient INPs in different temperature regimes. For example, black carbon, organic acids, humic like substances, and ammonium sulphate are found to be inefficient at temperatures above $-38$°C (Hoose and Möhler, 2012). Mineral dust becomes effective as INPs at temperatures of $-20$°C

and lower and is very common in the atmosphere. Biological aerosol particles are already active as INPs at temperatures above $-15$°C, with some already showing ice activity even above $-5$°C (Pummer et al., 2015). Biological INPs originate from the biosphere, typically connected to microorganisms such as bacteria, fungi, pollen and lichen (Pummer et al., 2015; Kanji et al., 2017). Their contribution to the atmosphere is expected to be limited in winter times, due to a less active biosphere and possibly due to ground and vegetation being covered by snow.

Given there is such an INP available and is immersed in a droplet at a temperature where the INP is active, the droplet freezes immediately. Depending mainly on temperature, supersaturation, and turbulence during the growth period of ice crystals, diverse crystal shapes are formed (Bailey and Hallett, 2009). For example at temperatures of $-3$ to $-10\,°$C, prolate shapes like needles, columns, and prisms dominate. At temperatures between $-10\,°$C and $-20\,°$C, oblate shapes like dendrites and plates dominate (Libbrecht, 2008). The formed pristine ice crystals then interact with the available cloud droplets and ice crystals,

which changes their shape during their way through the cloud. Therefore, the shape and size of a falling snow crystal holds important information about its formation process and life cycle. Complex snowflakes are formed through the aggregation of multiple ice crystals. Round shapes are formed by a process called riming, where water droplets freeze onto the crystal (Ávila et al., 2009; Maahn et al., 2024). Recent studies show that INP concentrations can vary between different air masses and that this has an impact on the characteristics of clouds (Radenz et al., 2021; Griesche et al., 2021). Radenz et al. (2021)

found that clouds in an environment with a low availability of INPs (as in Punta Arenas, Chile) tend to form less ice at the same temperatures clouds would do in an INP-loaded environment as in Leipzig, Germany, or Limassol, Cyprus. Griesche et al. (2021) found similar results for the Arctic. There, a higher fraction of ice-containing clouds was found compared to the mid-latitudes, especially at temperatures between $-3\,°$C and $-10\,°$C when the clouds were thermodynamically connected to the presumably INP-enriched summertime marine boundary layer.

Slightly supercooled stratiform clouds are frequently found over the Swiss Plateau in Central Europe during the so-called Bise situation. The Bise is a regional, persistent northeasterly or easterly wind over the Swiss Plateau, favouring the formation





of these clouds in wintertime. In case of the presence of a high-pressure system north of the Alps and a low-pressure system in Eastern Europe, the north-south surface pressure gradient is increased. The Bise is also a result of the local orography of the Swiss Plateau. Located in a long valley that gets narrower from northeast towards southwest, the Swiss Plateau is located between the Jura and the Alpine mountains. Therefore, a channeling of the air flow from the northeast leads to higher wind speeds in the area, typically accompanied by low temperatures (Wanner and Furger, 1990). Most of the times during a Bise situation, slightly supercooled stratiform clouds form over the Swiss Plateau and the neighbouring regions to the north and northeast. This makes it an ideal location to learn about ice formation at slightly supercooled conditions. Precipitation in these latitudes is mainly formed through the ice phase (Mülmenstädt et al., 2015). Therefore, INPs active at temperatures between $0\,°C$ and $-10\,°C$, which is a typical Bise cloud minimum temperature, are required to form precipitation. Precipitation would be an important factor to remove moisture from the cloud and to reduce its longevity. However, observations show that only every tenth stratiform cloud event over the Swiss Plateau at temperatures around $-10°C$ forms precipitation (Scherrer and Appenzeller, 2014; Granwehr , 2022). Comparable clouds over Germany are expected to contain ice much more frequently. Kanitz et al. (2011) found that one third of the clouds with minimum temperatures around $-10°C$ contain ice in Leipzig, Germany. This contrast could explain why models often struggle representing the longevity of the so-called Bise clouds and overestimate the amount of precipitation during Bise situations (Ohneiser et al., 2025a). Reasons for the lack of ice and precipitation in the supercooled clouds over the Swiss Plateau remain unclear and complicate the regional weather forecasts. Further research is required to understand the lack in ice formation in low supercooled clouds over the Swiss Plateau, to improve local weather models and precipitation forecasts. One attempt has recently been performed in the framework of the CLOUDLAB project, which used artificial cloud seeding to improve the understanding of the ice formation process in the supercooled Bise cloud layers (Henneberger et al., 2023).

This publication aims at contributing to the understanding of the complex topic of regional INP variability by means of a combination of INP observations at two different sites located along the typical trajectory of the Bise air masses, namely Hohenpeißenberg, Germany (HPB, 47.80°N, 11.01°E, 977 m a.s.l. upwind station) and Eriswil, Switzerland (Eri, 47.07°N, 7.87°E, 921 m a.s.l., downwind station). This is supported by INP measurements from Melpitz, Germany (51.53°N, 12.94°E, 86 m a.s.l.), which is a measurement station located at a rural area in a flat terrain with fields and meadows in the nearby vicinity, and by the incorporation of in-situ observations at a further field site over the Swiss plateau.

## 2 Starting hypothesis

The central hypothesis for this study is illustrated in Fig. 1a). We hypothesize that an observed lack of ice formation in supercooled low-level stratiform liquid clouds over Eriswil is due to a lack of INPs in the temperature range in which the clouds form. Further, we hypothesize that, under prevailing northeasterly winds, the few present INPs are activated and removed via precipitating from the stratus cloud deck, and that this would happen over some distance. The hypothesis was formed based on the observation, that during a cold Bise cloud at Eriswil there is typically no significant precipitation, but still, ice crystals form sporadically and can be observed at the ground (illustrated in Fig. 1a)). In our study region, during northeasterly winds,



air masses originate typically from eastern and northeastern Europe. During winter, a high amount of cloud condensation nuclei (CCN) resulting from industry and traffic is transported in these air masses towards Switzerland. However, we expected INP sources active at temperatures around $-10°C$ to be limited, with biological particles, and therewith INP active at these temperatures, being rare in the cold and snowy eastern Europe during winter times. Following the hypothesis, no significant INP concentrations would be left in the respective temperature regime of the supercooled cloud between 0 and $-10°C$ at a downwind

site as Eriswil, while somewhat higher INP concentrations may be expected at an upwind side as in Hohenpeißenberg. To generate the observed sporadic fall of ice crystals, we propose the following: On top of the stratus cloud, there is typically a strong temperature inversion. In the air above the inversion, in the free troposphere, there is still an INP reservoir available, relevant for the temperature range within the stratus clouds. Via turbulence, the INPs can be entrained into the liquid water cloud and form ice sporadically. A warm Bise situation (temperatures above freezing) or a clear sky situation would not decrease the

INP concentration in the PBL significantly.

For investigating this hypothesis, the air masses of two research sites often connected by air flow were sampled. The upwind site was at Hohenpeißenberg, Germany (coordinates: 47.80°N, 11.01°E, 945 m a.s.l.), and the downwind site at Eriswil, Switzerland (coordinates: 47.07°N, 7.87°E, 921 m a.s.l.). Figure 1b) provides a map showing the locations of Eriswil in central Switzerland and Hohenpeißenberg in southern Germany for spatial context. In between both stations there is a distance of

105 approximately 250 km. Measurements within the planetary boundary layer inside and outside of clouds reveal insight into the activation and removal of INPs. These measurements (wihtin PBL) in air masses at quasi neighbouring sites allow conclusions about the INP development between these sites if no INP sources are in between. In addition, INP measurements were operationally done at a further upwind site in Melpitz, Germany (51.53°N, 12.94°E, 86 m a.s.l.), which is approximately 440 km towards the north-east of Hohenpeißenberg. The site should be affected by similar air masses coming from north easterly

directions even though the trajectories might not have completely passed Melpitz before Hohenpeißenberg and Eriswil.





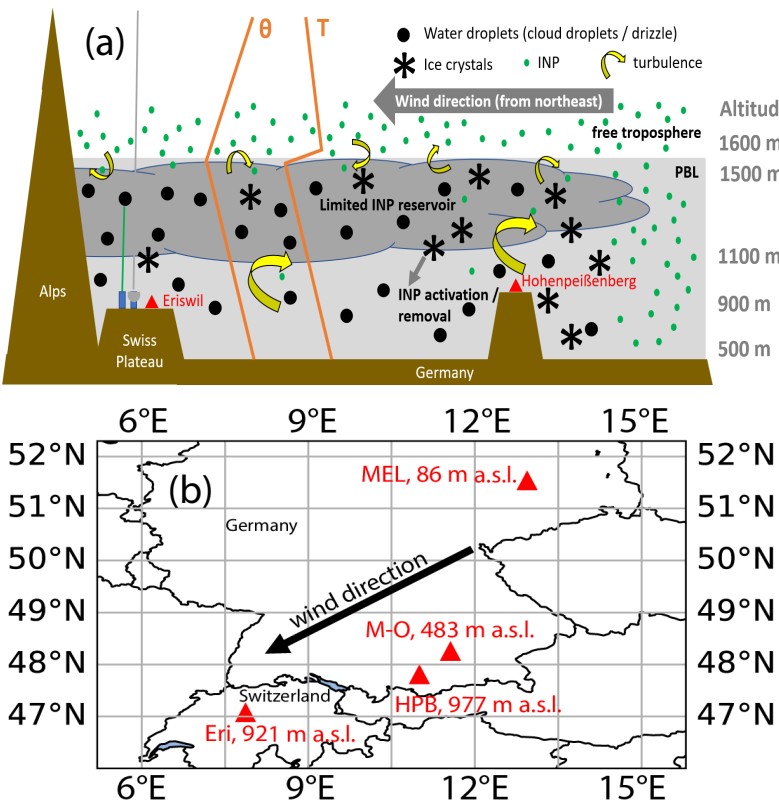

**Figure 1.** a) Sketch of the hypothesis of an experimental scenario that the INPs get lost within the supercooled stratus cloud after a short time. b) Map of the locations of Melpitz (MEL), München-Oberschleißheim (M-O), Hohenpeißenberg and Eriswil.

# 3 Data and methods

## 3.1 Experimental setup and data

The two measurement sites in Eriswil and Hohenpeißenberg were equipped with a set of remote-sensing and in-situ devices. The setup of the site in Eriswil is described in more detail in Ohneiser et al. (2025a). Table 1 provides a list of instruments that were used in this study.



**Table 1.** Details on the measurement instruments used in this study.

| Idx | Instrument (reference) | Frequency $\nu$ Wavelength $\lambda$ | Quantity |
|---|---|---|---|
| 1 | HATPRO G5 Microwave radiometer (TROPOS) (Rose et al., 2005) | $\nu$ = 22.24–31.4 GHz $\nu$ = 51.0–58.0 GHz | liquid water path (LWP), integrated water vapor, brightness temperatures |
| 2 | CHM-15kx Ceilometer (TROPOS) (Wiegner and Geiß, 2012) | $\lambda$=1064 nm | attenuated backscatter cf. |
| 3 | 2DVD two-dimensional video disdrometer (Schönhuber et al., 2008) | white light | particle number concentration, hydrometeor shape, type, size, oblateness |
| 4 | RPG94 FMCW-DP cloud radar, vertically pointing (RPG, 2024) | $\nu$=94 GHz | reflectivity, Doppler velocity, slanted linear depolarization ratio |
| 5 | VISSS Video In Situ Snowfall Sensor (Maahn et al., 2024) | $\lambda$=530 nm | particle number concentration, hydrometeor shape, type, size, oblateness, aspect ratio |
| 6 | Digitel DPA-14 low volume aerosol sampler | – | INP concentration via offline droplet freezing techniques |
| 7 | Windsond S1H3 (Bessardon et al., 2019) | – | temperature, pressure, relative humidity, wind speed, wind direction |

Measurements were done in the framework of the CLOUDLAB (Henneberger et al., 2023) and PolarCAP (Ohneiser et al., 2025a) campaigns. In addition, INP sampling was conducted in Eriswil and additionally at the German Weather Service (Deutscher Wetterdienst, DWD) in Hohenpeißenberg, using a low volume aerosol filter sampler (DIGITEL, DPA14 LVS) with a PM10 inlet at both sites. The filter magazines for the instruments at both sites were prepared simultaneously prior to
shipment to the sites. This was done in a laminar flow hood at TROPOS to eliminate possible contamination on the filters. 47 mm polycarbonate filters with a pore size of 0.4 $\mu$m were used. A filter sample typically was collected for 12 hours, with a flow rate of 25 L/min at ambient conditions resulting in a typical sampling volume of approximately 18000 STP L. During a Bise situation, the sampler was set to run continuously with an automatic filter change typically scheduled for 8 UTC and 20 UTC (see Ohneiser et al., 2025b). Each sampled filter was stored in a separate Petri dish at −20°C after sampling until off-line
analysis was done at the TROPOS ice-laboratory. Field blind filters were also taken by inserting filters into the sampler, but without sucking air through them, and by subsequently treating these field blinds similar to sampled filters. The same type of sampler but with an inlet for total suspended particles collected particles on polycarbonate filters (pore size of of 0.8 $\mu$m) for





INP analysis operationally at Melpitz. A filter is taken every third day for 24 hours with a flow rate of about 22 L/min. A field blind is taken every 9 days. Filter handling and evaluation were the same as for filters collected at Eri and HPB.

Also, LACROS (Leipzig Aerosol and Clouds Remote Observations System Radenz et al., 2021; Ohneiser et al., 2025a) data of the RPG94 Cloud radar, VISSS, 2DVD, HATPRO, and lidar PollyXT were used. In addition, windsondes were launched in Eriswil and radiosonde profile information for Hohenpeißenberg was taken from Uni Wyoming (2024) for the Munich station. Surface temperature information for Hohenpeißenberg was taken from the CDC (Climate Data Center from DWD, CDC , 2024).

## 3.2 INP measurements

Immersion freezing measurements were performed in the TROPOS ice-laboratory, using the two measurement devices Leipzig Ice Nucleation Array (LINA) and Ice Nucleation Droplet Array (INDA) (Hartmann et al., 2019). Both devices have been used in more than 20 studies in the past years, among them a comparison study done by Lacher et al. (2024), in which their results agreed well with those of other similar devices.

In short, INPs were washed off a filter sample by shaking the filter in 3 ml of ultra-pure water. For a LINA measurement, from this particle suspension, 90 droplets with a volume of $1\,\mu$l were then set onto a hydrophobic glass slide which itself sat on a Peltier element. On the glass slide, there was a grid which then again was covered with a second glass slide to separate the droplets from each other and the environment. The Peltier element cooled the droplets with a rate of 1 K/min. The setup was illuminated from above, and a picture was taken every 6 s (i.e., every 0.1 K) with a camera that was also installed above. 145 Frozen droplets were discriminated from non-frozen droplets by a custom, Python-based image analysis algorithm. The result from such a measurement is a curve of frozen fractions $f_{\mathrm{ice}}$ (i.e., number of frozen droplets / total number of droplets) over temperature $T$.

The remaining suspension was diluted further by adding 3.5 ml ultra-pure water in order to create enough suspension to fill the wells of a PCR plate with $50\,\mu$l droplets to be used with the INDA device. PCR plates with 96 wells were used, and most 150 of the time, two samples were examined simultaneously by filling 48 tubes with one sample and the other 48 tubes with the other (for Melpitz samples a full 96 well PCR plate per was used for each sample). After sealing the PCR tray with a foil, it was immersed in an ethanol bath such that the liquid in the bath was above the liquid level in the tubes. The bath was then cooled at 1 K/min. During cooling, the ethanol level was held constant with a custom-designed bath leveler similar to David et al. (2019). A light in the bath illuminated the PCR plate from below, and a camera was installed above and coupled with the 155 cryostat in order to take pictures in 0.1 K intervals. Again, frozen droplets were discriminated from non-frozen droplets with a custom, Python-based image analysis algorithm yielding $f_{\mathrm{ice}}(T)$.

Afterwards, the PCR plates were heated to 90°C for 30 min, cooled down and measured again with INDA to destroy heat-labile INPs, which is a proxy for the fraction of biological INPs.

From all frozen fraction curves, INP concentrations were then calculated based on the assumption of a Poisson distribution 160 of INPs via the examined droplets or wells, following Vali (1971). A more thorough description of the INP measurements can be found e.g., in Hartmann et al. (2019), Gong et al. (2022) or Sze et al. (2023).





### 3.3 Hybrid Single-Particle Lagrangian Integrated Trajectory Model – HYSPLIT

HYSPLIT (Hybrid Single-Particle Lagrangian Integrated Trajectory Model, Stein et al., 2015) is a model used to calculate trajectories, origin of air parcels, and dispersion. The model is freely available at HYSPLIT (2024). GDAS (Global Data
Assimilation System, 1°-spatial resolution) meteorological data in terms of air pressure, temperature, humidity, wind speed, wind direction, or precipitation are required. The HYSPLIT model uses a Lagrangian approach (applied in the study) as well as a Eulerian approach. The Lagrangian approach is used with a moving frame of reference for the advection and diffusion calculations. The Eulerian approach uses a fixed three-dimensional grid as a frame of reference to compute pollutant air concentrations. The trajectories were calculated twice per measurement day at 2 and 14 UTC at 500, 1900 and 4500 m a.g.l.
(above ground level) as 72–h backward trajectories.

### 4 Observations

INP sampling was done side by side with the CLOUDLAB and PolarCAP projects. During winter 2023/2024, there were two periods with a multi-day northeasterly wind (Bise) situation. The first period started on 6 Jan 2024 and ended on 12 Jan 2024. The second period started on 26 Feb 2024 and ended on 1 Mar 2024. In the first period, measurements with supercooled
conditions at the stratus cloud top were found, in the second period no supercooled conditions were observed.

#### 4.1 Overview over the two measurement periods

Figure 2a) presents the temperature evolution during the cold measurement period (6–12 January 2024) as recorded at the field site of Eriswil and for HPB data from the DWD CDC portal (CDC , 2024). Throughout this period, temperatures remain consistently below freezing at ground level, with the exception of a short interval on 6 January 2024 in Eriswil. Especially
on 8 and 9 January 2024, temperatures range between −5 °C and −10 °C. A notable temperature increase is observed at Hohenpeißenberg on 10 January 2024, where temperatures increase sharply from −10 °C to 0 °C within a few hours.

Figure 2c) displays vertical profiles of temperature for Eriswil for the same time period as Fig 2a), showing the presence of a temperature inversion typically occurring between 100 and 1000 m above ground. Stratus clouds are located between 200 and 1500 m above ground level. On 9 and 11 January, the inversion strength reaches nearly 10 K, resulting in temperatures around
185 0 °C in parts of the lower atmosphere at the inversion top.

In Fig. 2e) the most remarkable change in temperatures happened in Hohenpeißenberg (represented by radiosonde launches in München-Oberschleißheim) between 10 Jan 2024, 00 UTC and 12 UTC. The inversion decreases in altitude by a few hundred meters which results in a temperature increase of around 10 K in Hohenpeißenberg, as already mentioned above (Fig. 2a). Due to that, Hohenpeißenberg is no longer situated in the PBL but in the free troposphere.
During the warm measurement period (Fig. 2b), surface temperatures at both Eriswil and Hohenpeißenberg remain consistently above freezing. Temperatures generally stay below 6 °C, except for Hohenpeißenberg, where they occasionally increase to 10 °C, especially on 26 February 2024.





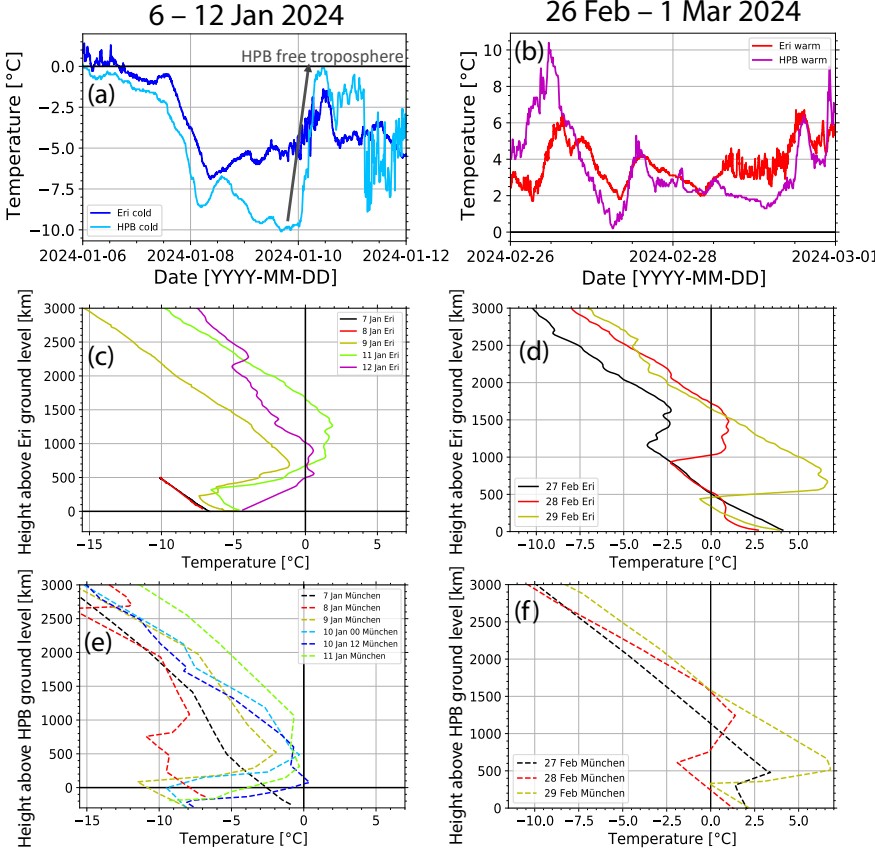

**Figure 2.** Temperature evolution during both measurement periods. a) Surface temperature during cold period (6–12 January 2024, "Eri" for Eriswil, "HPB" for Hohenpeißenberg). The transition time when the inversion has lowered and Hohenpeißenberg is no longer in the boundary layer but in the free troposphere is indicated with a gray arrow. b) Surface temperature evolution for the warm period (26 Feb 2024 to 1 Mar 2024). c) Temperature profiles for the cold period in Eriswil. d) Temperature profiles for the warm period in Eriswil. e) and f) Temperature profiles for Hohenpeißenberg for the cold and warm periods, respectively, represented by München-Oberschleißheim data (CDC , 2024). Eriswil surface data are observed by LACROS instrumentation at the field site in Eriswil. Hohenpeißenberg profile data are from Uni Wyoming  (2024). Eriswil profile data are from radiosonde launches at Eriswil. Note that a.g.l. relates to the height of 921 m for Eriswil and 977 m for Hohenpeißenberg.

The vertical temperature profiles for the warm period are shown in Fig. 2d) and f). Temperature inversions are present during this period as well. The surface temperatures ranges from $2\,°C$ to $4\,°C$, decreasing to around 0 to $-3\,°C$ at the lower boundary
of the inversion. The upper boundary of the inversion typically remains above freezing temperatures.

Figure 3 provides an overview of radar reflectivity at Eriswil during the two measurement periods. Figure 3a) shows the cold period. Frontal systems produce vertically thick clouds on 6 and 7 January 2024, as well as in parts on 8 and 9 January 2024. Nevertheless, the supercooled low-level liquid Bise cloud layer remains persistent, even during these times (as it can be seen





from the increased radar reflectivity between 0 and 1 km altitude). From noon on 8 January 2024 until midnight on 12 January
2024, the Bise cloud persists continuously, with only brief interruptions during the night from 9 to 10 January 2024. During
this time, the cloud top slowly descends from approximately 1400 m to between 100 and 200 m above ground. INP filters were
sampled throughout this period, although there were technical issues at both measurement sites, as indicated in the annotations
above Figs. 3a) and b) (white gaps between the green lines).

Figure 3b) shows the reflectivity during the warm period. The Bise cloud formed on 27 February 2024 and persisted until the
205 evening of 29 February 2024. During this warm Bise event, temperatures were mostly above freezing, both at the surface and
within the cloud. Filter sampling was successfully conducted throughout the entire warm Bise period.

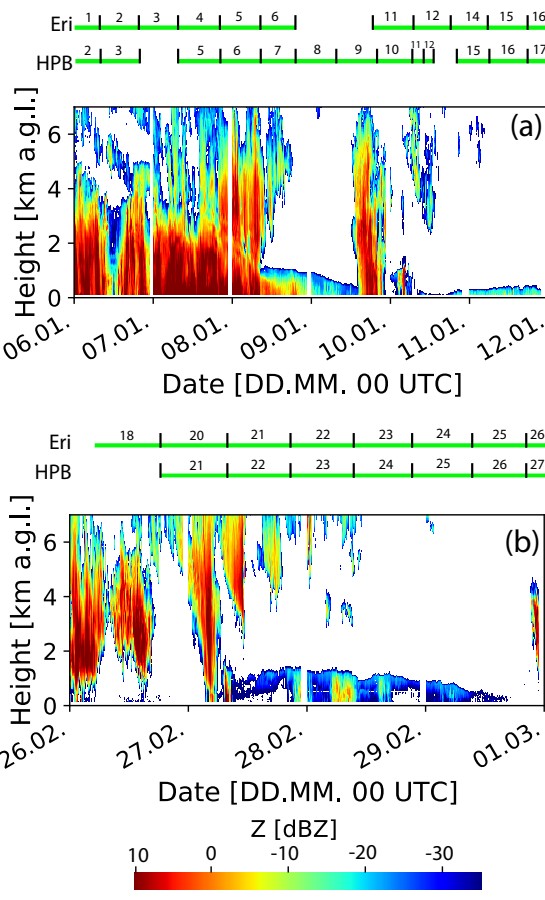

**Figure 3.** Radar reflectivity overview for the two periods (RPG94 cloud radar) for 6–11 January 2024 (cold period in a) and for 26–29
February 2024 (warm period in b) at Eriswil. In a) and b) the green lines on top indicate the measurement time period for each of the filters
and the number of the filter sample. Note, that the individual numbers for Eri and HPB are not necessarily synchronous.



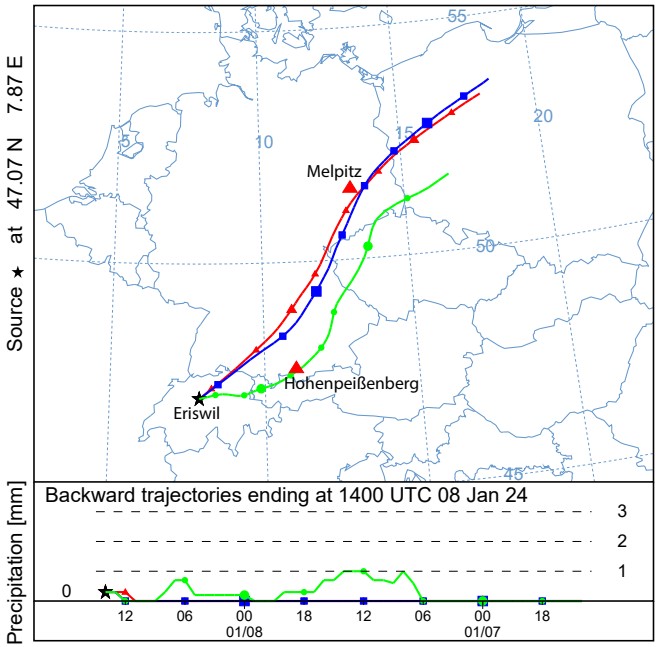

**Figure 4.** HYSPLIT backward trajectories ending at Eriswil on 8 Jan 2024, at 14 UTC at 100, 250, 1000 m a.g.l., shown in red, blue, green, respectively.

Figure 4, and Figs. S5, S6, and S7 in the supplement show the backward trajectories ending at Eriswil at 14 UTC on 8, 9, 10, and 11 January 2024, respectively. In all cases it can be seen, that the backward trajectories move very close to Hohenpeißenberg and especially on 8 January 2024 the trajectories are aligned with air masses blowing first over Melpitz, approximately 18 h later over HBP and another 12 h later over Eriswil. It can be assumed that the air mass is the same over all places coming roughly from the north-east. Along the trajectory there was only little precipitation.

## 4.2 INP measurements during cold and warm Bise in Eriswil and Hohenpeißenberg

Results from INP measurements are shown in Fig. 5, covering three different scenarios, each in a separate row. In each column, the leftmost panels Fig. 5a), d) and g) show a sketch of the respective situation, while all other panels show INP concentration spectra. INP spectra from Eriswil are always shown together with data from Hohenpeißenberg taken 12 hours earlier. During a Bise situation, the air mass moves towards Eriswil from northeasterly directions. We therefore assume that a comparable air mass also moved over Melpitz before. Therefore, INP measurements from Melpitz that correspond best in time are also added to the respective panels of Fig. 5. For Melpitz, there was no snow coverage on the ground at the time of the here shown measurements. Concerning Eriswil, as mentioned before artificial cloud seeding was done with a drone in supercooled Bise cloud layers which could in principle be a source of INP. However, we want to note that we checked the entire dataset for





signals from artificial seeding in the INP measurements. No evidence of cloud seeding particles on the ground was observed throughout the whole campaign.

Figure 5b) and c) each show the INP measurements at Eriswil and Hohenpeißenberg for the situation when both sites are
225 within the stratus cloud. The cloud top temperatures are below 0°C and overall the temperatures in these clouds range between 0 °C and −10 °C. INP concentrations are very similar at Eriswil and Hohenpeißenberg. Heating causes a decrease in INP concentrations, indicating a fraction of biological INPs, however, not as pronounced as for the warm Bise situation in Fig. 5h) and i). Two days earlier in Melpitz, the INP concentrations are much higher compared to the other two stations.

In the middle row, Fig. 5e) and f) show INP measurements for a time when Eriswil is located in the upper part of the
230 Bise cloud within the PBL, equivalent to the measurements shown in Fig. 5b) and c). Hohenpeißenberg, however, is slightly above the height level of the temperature inversion and therefore within the free troposphere after the inversion has lowered in altitude (see Fig. 2). The lowering of the inversion layer heights at Hohenpeißenberg occurs at 10 Jan 2024 around 3 UTC. Therefore, Hohenpeißenberg INP data in Fig. 5e) stems from air collected from both the PBL and the free troposphere. In the time period from 10 Jan, 8 pm to 11 Jan, 8 am shown in Fig. 5f) the measurement at HPB was performed entirely within the
235 free troposphere. Already in Fig. 5e), the INP spectrum from Eriswil is clearly below that from HPB. That, however, results from a lowering of the INP concentrations at Eriswil, visible when comparing Fig. 5c) to Fig. 5e). For Fig. 5f), when HPB was entirely in the free troposphere, a clear increase in INP concentrations at HPB is seen. For Fig. 5e) and f) the fraction of heat-labile biological INPs is negligible at Eriswil and therewith even lower than before. But when HPB is in the free troposphere, there is a high heat-labile INP fraction. For Melpitz, which is always located in the PBL, INP spectra are always at the high
end of observed INP concentrations. However, in Fig. 5f) INP concentrations from HPB start to resemble those from Melpitz. Interestingly, these INP concentrations for HPB show strong similarity to the warm Bise situation in Fig. 5h) and i).

The described results may indicate the presence of similar INP concentrations over a wide area, possibly emitted from snow-free regions upwind of HPB and even further upwind from Melpitz. Potentially these emissions were present sufficiently long to reach the free troposphere. In this context, it is interesting to note that Melpitz and a broad region north of Melpitz, extending
over most of North-Germany and Poland, was not covered by clouds (see example from 8 Jan 2024 in the supplement). Once a Bise cloud forms, INPs are removed quickly from the cloud, resulting in the low INP concentrations presented above.

In February, a warm Bise situation is observed. INP spectra are shown in Fig. 5h) and i). Both Eriswil and HPB are within the stratus cloud and the cloud-top temperatures are not supercooled. INP concentrations at both locations are generally higher than during the cold Bise, and very similar to the respective values measured at Melpitz. The higher temperatures during the
250 warm Bise do not allow for INP activation in clouds. Still, this similarity between the three locations is remarkable, as they are several hundred kilometers apart. Concerning heat-labile biological INPs, while their fraction is typically very low at Eriswil and HPB during the cold Bise situation, during the warm Bise situation we observe a much higher heat-labile fraction of INPs at both stations at freezing temperatures above −15 °C. The presence of these heat-labile biological INPs is noteworthy so early in the year, particularly as they already were present in early January at Melpitz and at HPB when it was in the free
troposphere.



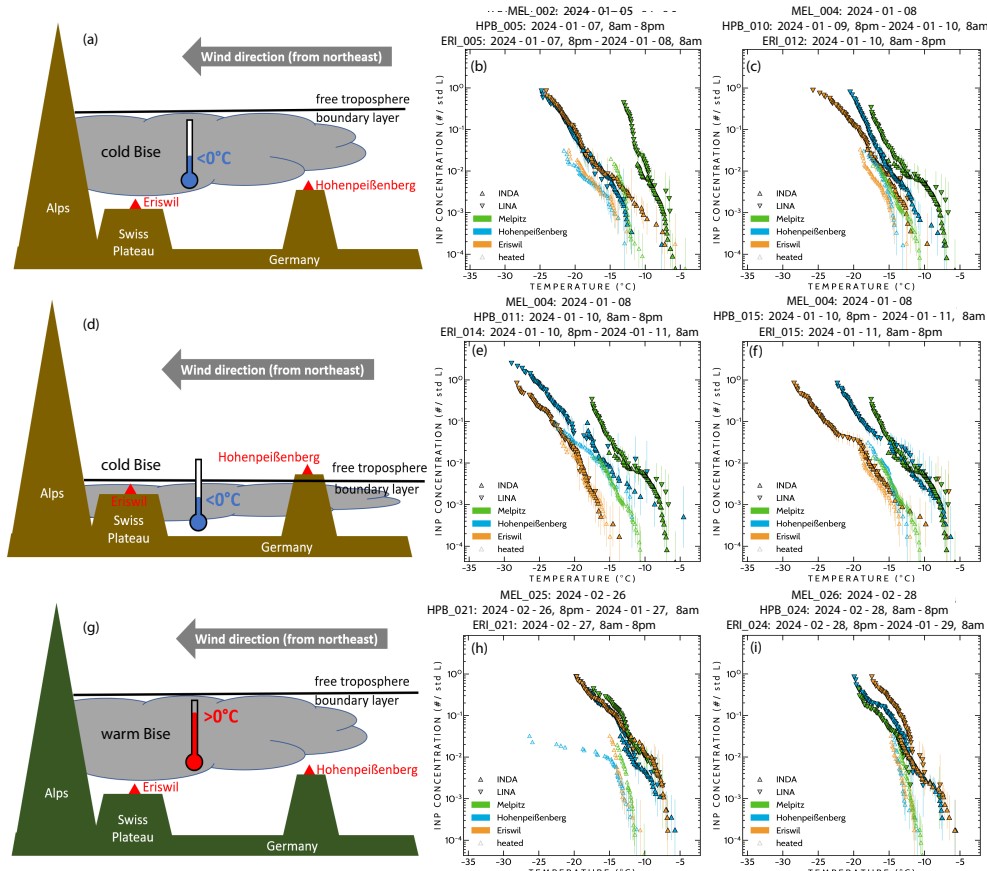

**Figure 5.** INP spectra for three different observed situations with a sketch of the situation in the leftmost column. Panels in the first row (a, b and c) refer to a situation when both Eri and HPB are within clouds during a cold Bise (cloud top temperatures below 0°C). The second row (d, e and f) depicts the situation of a cold Bise when Eri was within clouds but HPB was directly above clouds in the free troposphere. The third row (g, h and i) shows a warm Bise situation. INP spectra are always shown for Eri, HPB and Melpitz, with additional data for heated samples for Eri and HPB. For each scenario two case studies were chosen in order to show that the variability of the INP samples within the same scenario is relatively constant while the differences are rather significant for different scenarios.

## 4.3 Observations of ice crystals during Bise situation

Figure 6 presents a zoomed view of the RPG94 cloud radar reflectivity measurements covering the time period from 7 to 12 January 2024 from the surface to 1.5 km above ground level. Temperature profiles from two radiosonde launches are also provided. In the lowest 300 m, temperatures range from −5°C to −10°C, with warmer air observed above the inversion on 10

and 12 January 2024. The Bise cloud was affected by natural cloud seeding on 7 January which continues until approximately 8 January, 9 UTC (see Ohneiser et al., 2025a). Additionally, sedimenting ice crystals from upper-level cloud layers are visible in the evening of 9 January 2024.



The VISSS particle images (Fig. 6a1)–a3)) illustrate three periods of observations. The first period (Fig. 6a1), 8 January 2024, 3:23–3:59 UTC) shows ice crystals influenced by strong seeding from higher-altitude clouds, resulting in a diverse mixture of ice crystal shapes. Rimed particles, needles and aggregates are visible. The ice crystal number concentration (ICNC) at the surface observed by the 2DVD are around $10$–$100\,\mathrm{L^{-1}}$. During the second period (Fig. 6a2), 8 January 2024, 13:42–14:42 UTC), heavily rimed dendrites are observed. These particles initially form at cloud-top temperatures slightly colder than $-10°C$ and become heavily rimed as they fell through the approximately 1 km thick cloud layer, so they appear almost spherical. The ICNC was around $0.1\,\mathrm{L^{-1}}$ during this period. In the third period (Fig. 6a3), 11 January 2024, 13:00–14:57 UTC), ice crystals form at cloud-top temperatures around $-5°C$, resulting in needle-shaped particles observed at ground level. The ICNC is around $0.01\,\mathrm{L^{-1}}$ during this period.

The ICNC (ice crystal number concentration) at Eriswil between 8 and 12 Jan 2024 measured with 2DVD is shown in Fig. 6c). In the first half of 8 Jan 2024, seeder-feeder events caused the highest numbers of ICNC in the entire period of around 1 to $100\,\mathrm{ICNC\,L^{-1}}$. Without seeding, the ICNC decreased from 1 to 0.001 to $0.1\,\mathrm{L^{-1}}$. In the evening of 9 Jan 2024 another seeding event caused increased ICNC of up to $10\,\mathrm{L^{-1}}$. On 11 and 12 Jan 2024 the ICNC were typically between 0.0001 and $1\,\mathrm{L^{-1}}$. In general, a clear trend of decreasing ICNC during times without seeding is visible, from around 1 to 0.003. At the same time the Bise cloud thickness decreased from 1 km to 300 m and the temperatures in the cloud increased from around $-10°C$ to around $-5°C$.

The presented evolution of the observed ice crystals illustrates that light precipitation was recorded by VISSS throughout the observation period of the cold Bise situation, indicating continuous availability of at least a few INPs. Remote-sensing and in-situ measurements revealed, that concentrations of ice crystals were higher than the available INP concentrations. Measurable INP concentrations are only recorded at temperatures below the cloud-top temperatures (compare with the cold Bise INP measurements in Fig. 5b,c)) that prevail during the period of observation. This supports the hypothesis that INPs are entrained from the free troposphere via turbulence and afterwards immediately removed as they interact with the Bise cloud layer, leading to reduced availability of INPs downwind.




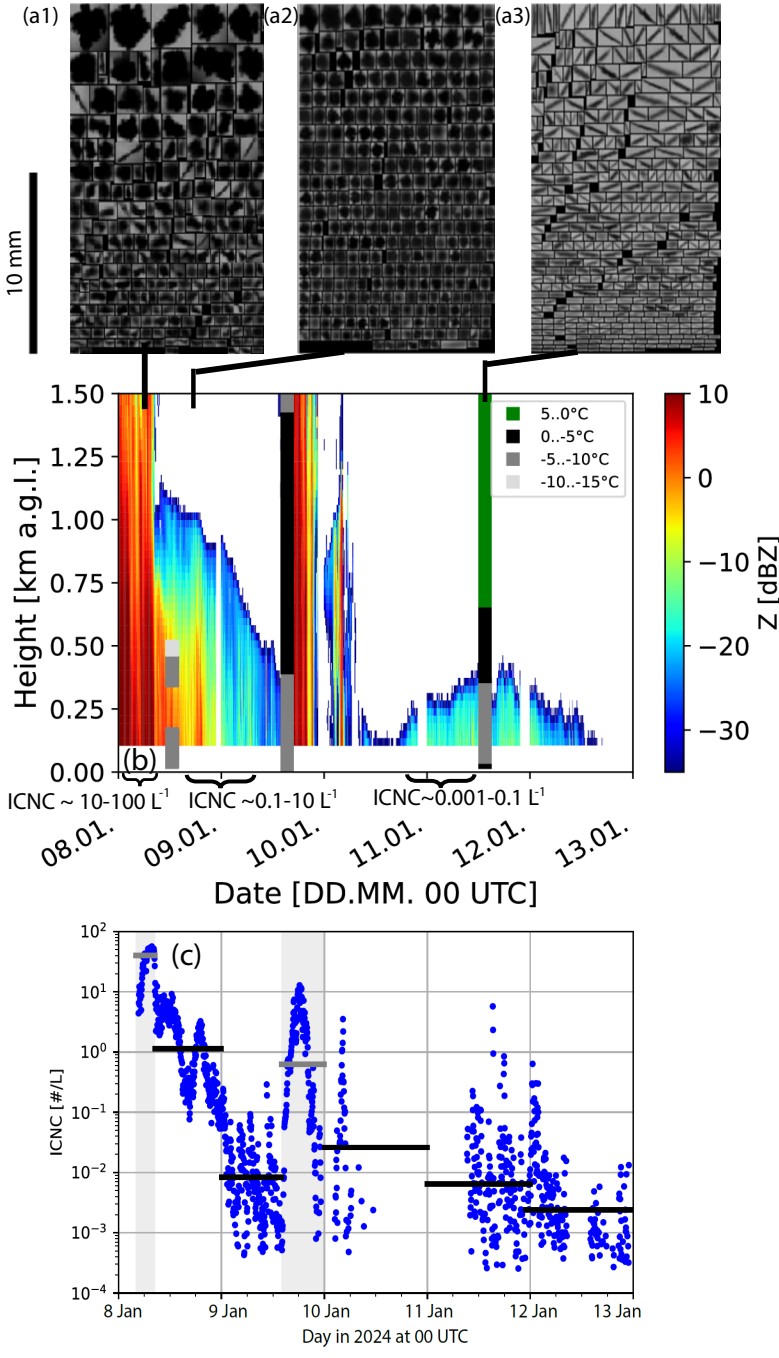

**Figure 6.** VISSS ice crystals for a1) 8 January 2024, 3:23–3:59 UTC a2) 8 January 2024, 13:42–14:42 UTC a3) 11 January 2024, 13:00–14:57 UTC b) RPG94 cloud radar reflectivity overview between 8 and 13 January 2024. Radiosonde data is shown as colored vertical bars in addition. c) ICNC between 8 and 13 Jan 2024 at Eriswil measured with the 2DVD. The gray and black horizontal lines are average ICNC during time periods with seeding and no seeding (typically 24-h averages), respectively.



## 5 Discussion and conclusion

In this study, it was shown that the INP concentration is generally very low in cold Bise-clouds in the PBL in the case of a cold Bise situation with stratus cloud cover. Under these conditions, both stations of Eriswil and Hohenpeißenberg showed similar INP spectra and generally a low biogenic INP fraction. Corresponding INP spectra from Melpitz showed much higher INP concentrations. For the case of a warm Bise, both stations showed similar INP characteristics, as well. However, the INP spectra were shifted towards higher temperatures and showed a larger fraction of biological INPs. With that, they resembled corresponding INP spectra from Melpitz. This general observation supports our hypothesis that INPs were removed from the PBL only during the cold Bise period. Interestingly, during the end of the cold period, Hohenpeißenberg was located in the free troposphere while Eriswil was still located in the boundary layer. While Eriswil showed a decrease of INP concentrations, the INP spectrum at Hohenpeißenberg showed higher INP concentrations and overall a strong similarity with the corresponding INP spectrum from Melpitz and also with the situation during the warm Bise period. Indeed, the INP concentration at Eriswil at the observed cloud-top temperatures would be too low in order to explain the sporadically but almost continuously observed ice crystals at the ground in Eriswil throughout the period. In the free troposphere, however, as it was measured in Hohen-peißenberg, there are still INPs available. These INP could have been mixed into the supercooled stratus cloud in the boundary layer via turbulence and form ice sporadically, analogously to the so-called dusty cirrus mechanism (Seifert et al., 2023) which support our hypothesis.

During the cold period when both stations were located in the boundary layer, we did not see significant differences in the INP spectra between Hohenpeißenberg and Eriswil. This does not support the second part of our hypothesis that INPs are lost on the way from Hohenpeißenberg towards Eriswil. More likely, the INPs are activated and removed quite fast once they are within the supercooled stratus cloud. However, within days, at Eriswil the INP spectrum during the cold Bise was still shifted towards colder temperatures. This could be a hint for ongoing INP removal. However, also a change in air mass, as indicated by the trajectories (see supplement Figs. S4, S5, S6, and S7) may be responsible for this change in INP concentrations. But in any case, the lack of INPs at Eriswil could explain the observation that not much ice is contained in the supercooled Bise clouds. Interestingly, the measurements in Melpitz are in contrast to the measurements in Hohenpeißenberg and Eriswil because for Melpitz, where PBL-coupled stratus clouds were absent in the investigated time period on 8 Jan 2024 (see supplement), there was no major difference in the observed INP concentrations between the examined periods of the cold and warm Bise.

In retrospective, not all atmospheric scenarios, which would have been required to thoroughly answer the question if the Bise clouds are responsible for the INP removal, could be observed during the campaign periods in winter 2023/2024. The observations did only allow for measurements in a cold Bise (cloudy, PBL and free troposphere) and warm Bise (cloudy). However, it would be interesting to have measurements in the case of a cold cloud-free Bise that goes along with clear sky in the PBL. It would allow to draw conclusions on the duration of the INP removal once the interaction with a supercooled cloud starts. In our study, we used INP measurements from the site of Melpitz to obtain a guess about how INP conditions might look like in absence of PBL-coupled supercooled stratus clouds. Nevertheless, we see high potential in conducting a future longer-lasting field experiment to evaluate the different INP and cloud scenarios in more detail, in order to answer whether short-scale INP

removal in a limited reservoir can lead to an observable lack of ice formation in supercooled liquid clouds. This would be of high relevance for increasing the understanding of the longevity of wintertime stratiform boundary layer cloud situations, such as the Bise.

**Data availability** Temperature profiles for Hohenpeißenberg for the cold and warm periods are represented by München-Oberschleißheim data (CDC , 2024). Hohenpeißenberg profile data are from Uni Wyoming (2024). Eriswil data are observed by LACROS instrumentation at the field site in Eriswil. The data can be found in (Ohneiser et al., 2025b). The dataset includes windsonde launches in Eriswil, surface temperature data from the HATPRO, 2DVD data, and radar reflectivity data from the RPG94. Details on the filter sample data and the INP data for HPB, Eri and Melpitz can be found in (Ohneiser et al., 2025b) as 330   well.

    **Author contributions** The measurements with LACROS in the frame PolarCAP were collected by KO, PS, AM, TG, VE, WS, and HG. The INP sampling at HPB was done by MH, KB and WT. The measurements in the frame of the Cloudlab project 335   were done by AM and JH. The filter samples were analyzed by AH, supported by MH and HW. The 2DVD data was analyzed by TG and KO. The manuscript was written by KO, MH, HW, and PS with support by all co-authors.

    **Financial support** Funding for this study was provided by the Deutsche Forschungsgemeinschaft (DFG, German Re-340   search Foundation) within the priority program SPP 2115 PROM via project numbers 408027490 (PolarCAP) and 408008112 (PICNICC, CORSIPP), by the European Union's Horizon Europe projects CleanCloud (grant no. 101137639), the European Union's Horizon 2020 research and innovation program (CLOUDLAB, grant agreement no. 101021272), the European Union's EU-HORIZON-WIDERA-2021 twinning program (BRACE-MY, grant no. 101079385) and the DFG project EMPOS (project number 516261703). The LACROS infrastructure received financial support via ACTRIS-D, which is funded by the Federal 345   Ministry of Education and Research of Germany under the funding code 01LK2001A.

    **Conflict of interest** The authors declare that they have no conflict of interest.

350

*Acknowledgements.* We acknowledge Matthias Bauer from Metek company for the great support with the Mira35 MBR5 and MBR7 cloud radars.



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
