# Peer review of "Ice-nucleating particle depletion in the wintertime boundary layer in the pre-Alpine region during stratus cloud conditions"

_EGUsphere, 2025_

## Referee Comment (RC1)

**Reviewer Comment**

October 7, 2025

The manuscript provides a good use case of connecting ice-nucleating particle (INP) measurements with synoptic conditions. The study is scientifically relevant and it will be interesting to see future more extensive measurements. The structure is clear and the used methods are explained well. The data is available online and in parts also published on Zenodo.

The manuscript needs some minor clarifications, but after adressing those, it will be a good and important addition to the journal.

**1 General comments**

1. Lines 109-110: Could you clarify what you mean with "the trajectories might not have completely passed Melpitz"? Trajectories are always associated with some uncertainties and also of course depend on the input data.

2. Lines 127-128: This is quite a large pore size with a high flow rate as well. Checking the literature (e.g., Cyrs et al. 2010), Nuclepore filters are to my knowledge typically used at lower flows to also make sure that particles are collected within the pores due to a longer residence time. Can you elaborate on this, how you make sure that 1) the Nuclepore filter does collect a representative aerosol sample from ambient air, and 2) what the pressure drop across the filter is? These Nuclepore filters are quite elastic and might be impacted negatively by a larger pressure difference across itself.

3. Line 152: Could you elaborate on the temperature measurement? It sounds like you use the temperature of the cryostat for your measurements. Did you ever check if there are inhomogeneities in the ethanol bath, which might lead to a temperature inhomogeneity within the PCR plates? This could also be answered by citing a relevant publication, if it exists.

4. Lines 163-170: Did you check the validity of the backward trajectories? I.e. calculation of the integration error using forward trajectories or investigating the change by moving the receptor some km in latitude and longitude direction? The trajectories are of course only an indication, but this indication could be increased in significance with additional quality checks.

5. Lines 251-255: I agree that heat-labile INPs can be seen as a proxy for biological INPs, but heat-labile INPs are not necessarily biological INPs and vice versa. Are you able to provide additional information to show that those are biological INPs, especially since you already measure them in early January, where the biological activity should be at a minima in Germany. I think you should either formulate it a little bit less strict, i.e. heat-labile is a proxy for biological, or you need to provide additional indications for the nature of the INPs.

6. Lines 281-285: You discuss here well, that you see a larger ice crystal number concentration compared to the concentration of INPs that can be activated at the given temperatures. Do you consider any secondary ice production processes, which could also be an explanation for what you observe?

7. Lines 289-291: See my comment above regarding the equivalence of biological and heat-labile INPs. Also try to be consistent with the use of biogenic and biological.

8. Line 302: Could you elaborate on what you consider a significant difference? Or is it just a visual check?

**2 Specific comments**

1. Line 34: "Mineral dust becomes effective as an INP at temperatures..." since mineral dust is used as a singular here.

2. Line 40: "Given there is such an INP available and is immersed in a droplet at a temperature where the INP is active, the droplet freezes immediately." The sentence reads not nicely to me. Maybe it could be reformulated in a bit simpler terms, i.e. "Upon the availability of an aerosol particle, which is immersed inside a supercooled droplet, at temperatures where the aerosol can act as an INP, the droplet freezes immediately." To me it makes most sense to discuss an aerosol particle, which can act as an INP below its activation temperature.

3. Lines 104-105: "The distance between the two stations is 250 km." Saying that a distance is between two points sounds wrong.

4. Line 105: Planetary boundary layer has already been defined as PBL above. Insight should also be insights, since you are not just getting a single insight from your measurements.

5. Line 106: wihtin PBL -> within PBL.

6. Lines 108-109: ...which is located approximately 440 km northeast of Hohenpeißenberg.

7. Line 122: Can you provide the conditions you refer to as standard? This can be widely different for different fields and groups and would provide needed information for comparison.

8. Line 145: Are there plans to make the Python code available online or publishing it on i.e. Zenodo?

9. Line 150: You discuss that you filled tubes with your sample, but then discuss wells of PCR plates. I assume you mean the same, therefore I would suggest to use wells throughout.

10. Lines 157-158: The sentence could be structured differently to make it a bit more clear that the heating removes the ability of heat-labile INPs to induce ice nucleation, i.e. "Afterwards, the PCR plates were heated to 90 °C for 30 min to remove the ability of heat-labile INPs, which are a proxy for the fraction of biological INPs, to induce ice nucleation."

11. Line 183-184: Above ground could be abbreviated via a.g.l., same as above ground level.

12. Line 201: Above ground could be abbreviated via a.g.l.

13. Line 226: Heating causes -> Heat treatment causes.

14. Line 258: Above ground level could be abbreviated via a.g.l.

15. Line 266: are around -> is around.

16. Line 272: You already abbreviate ICNC in the previous paragraph, therefore I would suggest that you do not need to do it here again.

17. Line 306: colder temperatures -> lower temperatures.

**2.1 Comments on tables and figures**

1. Table 1: There should be spaces around the equal sign in the third column.

2. Figure 3: Using the rainbow colormap has been the standard for remote sensing data, but I would recommend to use a colormap that first of all can be understood by readers with colour vision deficiencies. This is not given with the currently used colormap. In addition, the rainbow colormap has numerous other issues, for example the linearity, which is not given for example the lightness of the colormap (e.g., Kovesi 2015). This is especially pronounced when looking at the yellow or cyan color of the colormap. These are quite sharply separated from the other colors and therefore the data itself looks different due to the used colormap. I would recommend a perceptually uniform colormap, such as viridis or cividis.

3. Figure 5: This is a nice graphic and really helps the reader to understand the different scenarios. I do have some questions regarding the uncertainties. Could you elaborate on the calculation of the uncertainties for the different instrumentation? It seems like the relative uncertainty lowers for higher INP concentration, which I guess is related to using Poisson statistics? The second part is just a bit of curiosity. When looking at panel (h), the heated samples from Hohenpeißenberg shows a very similar slope compared to the other locations at around -12 °C, but then the curve almost falls flat with only around 10 freezing events. Do you have any idea, why that could be or what that could indicate?

4. Figure 6: Same comments apply for the used colormap as given for Figure 5.

**3 Technical comments**

1. Line 20: space is missing between number and unit.

2. Lines 23-24: space after author last name should be removed.

3. Line 24: space is missing between number and unit.

4. Line 31: space is missing between number and unit.

5. Lines 34-36: space is missing between number and unit.

6. Line 67: space is missing between number and unit.

7. Line 69: space is missing between number and unit.

8. Line 82: latin phrases should not be hyphenated.

9. Lines 79-81: coordinates need a degree sign and a space when naming the direction (e.g. 30° N, 25° E).

10. Line 92: space is missing between number and unit.

11. Line 94: space is missing between number and unit.

12. Lines 102-103: coordinates need a degree sign and a space when naming the direction (e.g. 30° N, 25° E).

13. Line 108: coordinates need a degree sign and a space when naming the direction (e.g. 30° N, 25° E).

14. Line 113: latin phrases should not be hyphenated.

15. Line 121: units should be written upright and not italic.

16. Line 122: units should be written utilizing exponents.

17. Line 124: space is missing between number and unit.

18. Line 127: units should be written upright and not italic.

19. Line 128: units should be written utilizing exponents.

20. Line 133: space after CDC should be removed.

21. Line 141: units should be written upright and not italic.

22. Line 143: units should be written utilizing exponents.

23. Line 149: units should be written upright and not italic.

24. Line 153: units should be written utilizing exponents.

25. Line 157: space is missing between number and unit.

26. Line 181: earlier you used a single unit when showing a range (i.e. line 42), this should be consistent.

27. Line 194: earlier you used a single unit when showing a range (i.e. line 42), this should be consistent.

28. Line 245: North-Germany is not capitalized and should probably be "northern Germany".

29. Line 259: earlier you used a single unit when showing a range (i.e. line 42), this should be consistent. In addition, there is a space missing between number and unit.

30. Line 268: space is missing between number and unit.

31. Lines 278-279: space is missing between number and unit.

32. Lines 280-281: latin phrases should not be hyphenated.

33. Line 334: Cloudlab is capitalized before (see line 75), this should be consistent throughout the manuscript.

**References**

Cyrs, W. D., D. A. Boysen, G. Casuccio, T. Lersch, and T. M. Peters (2010). "Nanoparticle collection efficiency of capillary pore membrane filters". In: *Journal of Aerosol Science* 41.7, pp. 655–664. DOI: 10.1016/j.jaerosci.2010.04.007.

Kovesi, P. (2015). "Good Colour Maps: How to Design Them". In: arXiv: 1509.03700 [cs.GR].

---

## Author Comment (AC1)

Color legend:
Question or comment by the reviewer
Our answer to a reviewer's question

**Reviewer Comment**

October 7, 2025

The manuscript provides a good use case of connecting ice-nucleating particle (INP) measurements with synoptic conditions. The study is scientifically relevant and it will be interesting to see future more extensive measurements. The structure is clear and the used methods are explained well. The data is available online and in parts also published on Zenodo. The manuscript needs some minor clarifications, but after adressing those, it will be a good and important addition to the journal.

**1 General comments**

1. Lines 109-110: Could you clarify what you mean with "the trajectories might not have completely passed Melpitz"? Trajectories are always associated with some uncertainties and also of course depend on the input data.

Thanks to reviewer #1 for hinting at this formulation. The formulation is inaccurate and confusing. What we wanted to say is that the uncertainties of the trajectories are large and we cannot be sure if indeed Melpitz was passed before HPB and Eri. In the updated version of the manuscript, we now leave out the second part of the sentence: "even though the trajectories might not have completely passed Melpitz before Hohenpeißenberg and Eriswil". We believe that this was the confusing part.

2. Lines 127-128: This is quite a large pore size with a high flow rate as well. Checking the literature (e.g., Cyrs et al. 2010), Nuclepore filters are to my knowledge typically used at lower flows to also make sure that particles are collected within the pores due to a longer residence time. Can you elaborate on this, how you make sure that 1) the Nuclepore filter does collect a representative aerosol sample from ambient air, and 2) what the pressure drop across the filter is? These Nuclepore filters are quite elastic and might be impacted negatively by a larger pressure difference across itself.

Regarding 1) We participated in an intercomparison campaign called "Puy de Dôme ICe Nucleation Intercomparison Campaign PICNIC" (Lacher et al., 2024) with the same instrumentation that was used for this manuscript. During PICNIC also filter sampling with differently sized pores and at different flow rates was compared and no significant influence of these parameters was found: "This shows that filters with a pore size of 800 nm and applied flow rate still have a sufficiently high collection efficiency for the majority of atmospheric INPs present during the PICNIC study. This is in agreement with Soo et al. (2016), who examined the collection efficiencies of a range of different filter materials and pore sizes for test particles with rather small sizes between 10 and 412 nm. They reported that the collection efficiency for polycarbonate filters with 800 nm pore sizes and the flow

rates used here (>11 L min−1) are above 97 % for all particles in the examined size range (10–412 nm)."(Lacher et al., 2024)

Regarding 2) The pressure was about 200 - 220 hPa.

3. Line 152: Could you elaborate on the temperature measurement? It sounds like you use the temperature of the cryostat for your measurements. Did you ever check if there are inhomogeneities in the ethanol bath, which might lead to a temperature inhomogeneity within the PCR plates? This could also be answered by citing a relevant publication, if it exists.

We do not have a dedicated publication about our DFA, but internally we of course performed numerous tests and established several calibration routines to ensure good data quality. We repeat a temperature calibration of our instruments yearly, and did not observe large changes over the past years. Temperatures used are then those resulting from the calibration.
Temperature homogeneity across a PCR plate is tested as part of that annual calibration. Inhomogeneities exist and the annual calibration relevant for the presented data showed a standard deviation of 0.3 K for the horizontal temperature distribution. But it should be noted that the inhomogeneity is not systematic, i.e., there is no persistent cold/warm spot in the same location across all measurements. This means that if the same sample is measured again, the freezing temperature of an individual well might change, but if looked at the full measured spectrum, the curves are essentially identical.

4. Lines 163-170: Did you check the validity of the backward trajectories? I.e. calculation of the integration error using forward trajectories or investigating the change by moving the receptor some km in latitude and longitude direction? The trajectories are of course only an indication, but this indication could be increased in significance with additional quality checks.

Yes, we calculated forward and backward trajectories and found a consistent situation of the trajectories arriving from northeasterly directions. The weather situation was also quite stable over many days and some of the calculated trajectories are also shown in the supplementary material. Motivated by the question raised by reviewer #2, we also checked arbitrary trajectories in a few km lat/lon distance of HPB and Eri and found similar results for the trajectories. So, we are confident that we have a robust result here.

5. Lines 251-255: I agree that heat-labile INPs can be seen as a proxy for biological INPs, but heat-labile INPs are not necessarily biological INPs and vice versa. Are you able to provide additional information to show that those are biological INPs, especially since you already measure them in early January, where the biological activity should be at a minima in Germany. I think you should either formulate it a little bit less strict, i.e. heat-labile is a proxy for biological, or you need to provide additional indications for the nature of the INPs.

We agree that heat-labile INPs and biological INPs are not synonymous, and that heat-labile INPs are merely a proxy for biological INP, which was stated in L157-L158 (line of the original manuscript), when the heat treatment was first mentioned. Since it was not our

intention to evoke a wrong impression, we adjusted our phrasing when the heat-labile INP were mentioned in the following locations (Line numbers refer to the revised manuscript):
L246: "Heat treatment causes a decrease in INP concentrations, indicating a fraction of heat-labile, probably biological INPs,[...]"
L257 : "[...] the fraction of heat-labile INPs is negligible at Eriswil"
L274:"Concerning the heat-labile, and therefore potentially biological INPs, [...]"
L281 – L282: "The presence of these potentially biological INPs is noteworthy [...]"
L319: "[...] showed similar INP spectra and generally a low biological INP fraction (indicated by the low fraction of heat-labile INP). "
L322: "[...] showed a larger fraction of heat-labile, potentially biological INPs."

The release of bioaerosol (including biological INP), is of course strongly linked with the biological activity and hence with the seasons in mid-latitudes, nevertheless can bioaerosol also be observed in winter. For example, while not as numerous as during summer, fungal spores can be also found during wintertime (Havis et al., 2023; Lagomarsino Oneto et al. 2020; Oliveira et al., 2009; Pady, S. M.,1957). Also the transport of microbes via aerosolized soil dust (Salawu-Rotimi et al., 2021) is a viable pathway during winter.

6. Lines 281-285: You discuss here well, that you see a larger ice crystal number concentration compared to the concentration of INPs that can be actived at the given temperatures. Do you consider any secondary ice production processes, which could also be an explanation for what you observe?

It is true that we only discussed the observed difference as an indication that the gap between INP and ICNC originates from free tropospheric INP intrusion. Therefore, we reformulated the whole paragraph (line 312-320): "Specifically, mean ICNC were between $10^{-3}$ and $10^{-2}$ L$^{-1}$ from 11 through 12 January 2024, when cloud-minimum temperatures were about -7°C. It is, however, remarkable that the INP observations of around $10^{-3}$ L$^{-1}$ at HPB, when this site was above the PBL (Fig. \ref{Fig:INP_contrast}e,f), were on a similar order as the ICNC concentrations observed during the same period at Eri. This supports the hypothesis that INPs are entrained from the free troposphere via turbulence and afterwards immediately removed as they interact with the Bise cloud layer, leading to reduced availability of INPs downwind. However, it must be noted that an ICNC concentration which is higher than the observed INP concentration can in principle also be a result of secondary ice formation processes (Korolev et al., 2020). Nevertheless, secondary ice formation processes generally lead to orders of magnitudes of increase in ICNC, which was, besides occasional peaks in the ICNC, not observed in the average ICNC values during the investigated time periods."

7. Lines 289-291: See my comment above regarding the equivalence of biological and heat-labile INPs. Also try to be consistent with the use of biogenic and biological.

We replaced "biogenic" in L327 with "biological"

8. Line 302: Could you elaborate on what you consider a significant difference? Or is it just a visual check?

In this instance it is simply based on the overlap of the confidence intervals in the relevant high temperature regime (>-15°C).

**2 Specific comments**

1. Line 34: "Mineral dust becomes effective as an INP at temperatures..." since mineral dust is used as a singular here.

Reviewer #1 is right. Rewriting the sentence is reasonable. We changed the sentence in the manuscript accordingly.

2. Line 40: "Given there is such an INP available and is immersed in a droplet at a temperature where the INP is active, the droplet freezes immediately." The sentence reads not nicely to me. Maybe it could be reformulated in a bit simpler terms, i.e. "Upon the availability of an aerosol particle, which is immersed inside a supercooled droplet, at temperatures where the aerosol can act as an INP, the droplet freezes immediately." To me it makes most sense to discuss an aerosol particle, which can act as an INP below its activation temperature.

We thank reviewer #1 for the suggestion of a more reasonable sentence structure. We changed the sentence in the manuscript (lines 42ff).

3. Lines 104-105: "The distance between the two stations is 250 km." Saying that a distance is between two points sounds wrong.

We changed the formulation in the manuscript.

4. Line 105: Planetary boundary layer has already been defined as PBL above. Insight should also be insights, since you are not just getting a single insight from your measurements.

This is correct. We use the introduced abbreviation PBL now.

5. Line 106: wihtin PBL -> within PBL.

It is corrected now.

6. Lines 108-109: ...which is located approximately 440 km northeast of Hohenpeißenberg.

We changed the sentence in the manuscript accordingly.

7. Line 122: Can you provide the conditions you refer to as standard? This can be widely different for different fields and groups and would provide needed information for comparison.

0°C and 1013 hPa, which was added to the respective line (line 131 of the revised manuscript).

8. Line 145: Are there plans to make the Python code available online or publishing it on i.e. Zenodo?

No. As this is quite specific to our set-up, we do not see the added value.

9. Line 150: You discuss that you filled tubes with your sample, but then discuss wells of PCR plates. I assume you mean the same, therefore I would suggest to use wells throughout.

Yes, we mean the same and now use "well" throughout the manuscript.

10. Lines 157-158: The sentence could be structured differently to make it a bit more clear that the heating removes the ability of heat-labile INPs to induce ice nucleation, i.e. "Afterwards, the PCR plates were heated to 90 °C for 30 min to remove the ability of heat-labile INPs, which are a proxy for the fraction of biological INPs, to induce ice nucleation."

We included the suggestion of reviewer #1 to reformulate the sentence (lines 167ff).

11. Line 183-184: Above ground could be abbreviated via a.g.l., same as above ground level.

We changed "above ground" to "a.g.l."

12. Line 201: Above ground could be abbreviated via a.g.l.

We changed "above ground" to "a.g.l.".

13. Line 226: Heating causes -> Heat treatment causes.

We changed "Heating causes" to "Heat treatment causes".

14. Line 258: Above ground level could be abbreviated via a.g.l.

We changed "above ground" to "a.g.l.".

15. Line 266: are around -> is around.

We changed "are around" to "is around".

16. Line 272: You already abbreviate ICNC in the previous paragraph, therefore I would suggest that you do not need to do it here again.

We removed the redefinition and only write ICNC now in the manuscript.

17. Line 306: colder temperatures -> lower temperatures.

We changed "colder temperatures" to "lower temperatures".

**2.1 Comments on tables and figures**

1. Table 1: There should be spaces around the equal sign in the third column.

We added spaces around the equal sign in the third column.

2. Figure 3: Using the rainbow colormap has been the standard for remote sensing data, but I would recommend to use a colormap that first of all can be understood by readers with colour vision deficiencies. This is not given with the currently used colormap. In addition, the rainbow colormap has numerous other issues, for example the linearity, which is not given for example the lightness of the colormap (e.g., Kovesi 2015). This is especially pronounced

when looking at the yellow or cyan color of the colormap. These are quite sharply separated from the other colors and therefore the data itself looks different due to the used colormap. I would recommend a perceptually uniform colormap, such as viridis or cividis.

We thank reviewer #1 for the comment. Indeed, the rainbow colormap is the standard for remote-sensing data. We acknowledge that readers with colour vision deficiencies would have difficulties with the interpretation of the figure and also the nonlinearity is an issue with the colormap. We changed the colormap of the figure towards viridis. It is updated in the manuscript.

3. Figure 5: This is a nice graphic and really helps the reader to understand the different scenarios. I do have some questions regarding the uncertainties. Could you elaborate on the calculation of the uncertainties for the different instrumentation? It seems like the relative uncertainty lowers for higher INP concentration, which I guess is related to using Poisson statistics? The second part is just a bit of curiosity. When looking at panel (h), the heated samples from Hohenpeißenberg shows a very similar slope compared to the other locations at around -12 °C, but then the curve almost falls flat with only around 10 freezing events. Do you have any idea, why that could be or what that could indicate?

Yes, this is the statistical uncertainty stemming from the Poisson distribution, calculated according to Agresti and Coull (1998). This method has become the de facto standard for statistical uncertainty in droplet freezing array measurements. Therefore, we did not include details in the manuscript but added a reference to the method in the caption of Figure 5:
 "Error bars represent the 95% confidence interval of the statistical uncertainty calculated after Agresti and Coull (1998)."

In general, such plateau-like features (here, approximately between -15°C and -26°C) indicate that no additional INPs can initiate freezing at that temperature. These features can occur when the sample consists of sufficiently distinct INP populations of a certain concentration or as an artifact of the dilution. In this case it is most likely an dilution artifact. It is difficult to answer with certainty why the heated Hohenpeißenberg sample shows this plateau but the samples from the other two locations do not.

4. Figure 6: Same comments apply for the used colormap as given for Figure 3.

Same as two comments before, we changed towards viridis colormap.

**3 Technical comments**

We thank reviewer #2 for carefully reading through the manuscript and for giving technical comments. We incorporated all of the following 33 comments.

1. Line 20: space is missing between number and unit.

2. Lines 23-24: space after author last name should be removed.

3. Line 24: space is missing between number and unit.

4. Line 31: space is missing between number and unit.

5. Lines 34-36: space is missing between number and unit.

6. Line 67: space is missing between number and unit.

7. Line 69: space is missing between number and unit.

8. Line 82: latin phrases should not be hyphenated.

9. Lines 79-81: coordinates need a degree sign and a space when naming the direction (e.g. 30° N, 25° E).

10. Line 92: space is missing between number and unit.

11. Line 94: space is missing between number and unit.

12. Lines 102-103: coordinates need a degree sign and a space when naming the direction (e.g. 30° N, 25° E).

13. Line 108: coordinates need a degree sign and a space when naming the direction (e.g. 30° N, 25° E).

14. Line 113: latin phrases should not be hyphenated.

15. Line 121: units should be written upright and not italic.

16. Line 122: units should be written utilizing exponents.

17. Line 124: space is missing between number and unit.

18. Line 127: units should be written upright and not italic.

19. Line 128: units should be written utilizing exponents.

20. Line 133: space after CDC should be removed.

21. Line 141: units should be written upright and not italic.

22. Line 143: units should be written utilizing exponents.

23. Line 149: units should be written upright and not italic.

24. Line 153: units should be written utilizing exponents.

25. Line 157: space is missing between number and unit.

26. Line 181: earlier you used a single unit when showing a range (i.e. line 42), this should be consistent.

27. Line 194: earlier you used a single unit when showing a range (i.e. line 42), this should be consistent.

28. Line 245: North-Germany is not capitalized and should probably be "northern Germany".

29. Line 259: earlier you used a single unit when showing a range (i.e. line 42), this should be consistent. In

addition, there is a space missing between number and unit.

30. Line 268: space is missing between number and unit.

31. Lines 278-279: space is missing between number and unit.

32. Lines 280-281: latin phrases should not be hyphenated.

33. Line 334: Cloudlab is capitalized before (see line 75), this should be consistent throughout the manuscript.

**References**

Agresti, A., & Coull, B. A. (1998). Approximate is Better than "Exact" for Interval Estimation of Binomial Proportions. The American Statistician, 52(2), 119–126. https://doi.org/10.1080/00031305.1998.10480550

Cyrs, W. D., D. A. Boysen, G. Casuccio, T. Lersch, and T. M. Peters (2010). "Nanoparticle collection efficiency of capillary pore membrane filters". In: Journal of Aerosol Science 41.7, pp. 655–664. doi: 10.1016/j.jaerosci.2010.04.007.

Havis, N.D., Kaczmarek, J., Jedryczka, M. et al. Spore dispersal patterns of the ascomycete fungus Ramularia collo-cygni and their influence on disease epidemics. Aerobiologia 39, 213–226 (2023). https://doi.org/10.1007/s10453-023-09787-6

Kovesi, P. (2015). "Good Colour Maps: How to Design Them". In: arXiv: 1509.03700 [cs.GR].

Lacher, L., Adams, M. P., Barry, K., Bertozzi, B., Bingemer, H., Boffo, C., Bras, Y., Büttner, N., Castarede, D., Cziczo, D. J., DeMott, P. J., Fösig, R., Goodell, M., Höhler, K., Hill, T. C. J., Jentzsch, C., Ladino, L. A., Levin, E. J. T., Mertes, S., Möhler, O., Moore, K. A., Murray, B. J., Nadolny, J., Pfeuffer, T., Picard, D., Ramírez-Romero, C., Ribeiro, M., Richter, S., Schrod, J., Sellegri, K., Stratmann, F., Swanson, B. E., Thomson, E. S., Wex, H., Wolf, M. J., and Freney, E.: The Puy de Dôme ICe Nucleation Intercomparison Campaign (PICNIC): comparison between online and offline methods in ambient air, Atmos. Chem. Phys., 24, 2651–2678, https://doi.org/10.5194/acp-24-2651-2024, 2024.

Lagomarsino Oneto, D., Golan, J., Mazzino, A., Pringle, & Seminara A., Timing of fungal spore release dictates survival during atmospheric transport, Proc. Natl. Acad. Sci. U.S.A. 117 (10) 5134-5143, https://doi.org/10.1073/pnas.1913752117  (2020).

Oliveira, M., Ribeiro, H., Delgado, J.L. et al. The effects of meteorological factors on airborne fungal spore concentration in two areas differing in urbanisation level. Int J Biometeorol 53, 61–73 (2009). https://doi.org/10.1007/s00484-008-0191-2

Pady, S. M. (1957). Quantitative Studies of Fungus Spores in the Air. Mycologia, 49(3), 339–353. https://doi.org/10.1080/00275514.1957.12024649

Salawu-Rotimi, A., Lebre, P.H., Vos, H.C. et al. Gone with the Wind: Microbial Communities Associated with Dust from Emissive Farmlands. Microb Ecol 82, 859–869 (2021). https://doi.org/10.1007/s00248-021-01717-8

Soo, J. C., Monaghan, K., Lee, T., Kashon, M., and Harper, M.: Air sampling filtration media: Collection efficiency for respirable size-selective sampling, Aerosol Sci. Tech., 50, 76–87, https://doi.org/10.1080/02786826.2015.1128525, 2016.